# Novel Feature Representation Strategies for Time Series Forecasting with Predicted Future Covariates

## Abstract

Accurate time series forecasting is a fundamental challenge in data science. Unlike traditional statistical methods, conventional machine learning models, such as RNNs and CNNs, use historical data consisting of previously measured variables including the forecast variable and all its covariates. However, in many applications, some of the covariates can be predicted with reasonable accuracy for the immediate future. We refer to such covariates as *predictable future covariates*. Note that the input may also contain some covariates that cannot be accurately predicted. We consider the problem of predicting water levels at a given location in a river or canal system using historical data and future covariates, some of which (precipitation, tide) may be predictable. In many applications, for some covariates of interest, it may be possible to use historical data or accurate predictions for the near future. Traditional methods to incorporate future predictable covariates have major limitations. The strategy of simply concatenating the future predicted covariates to the input vector is highly likely to miss the past-future connection. Another strategy that iteratively predicts one step at a time can end up with prediction error accumulation. We propose two novel feature representation strategies to solve those limitations – *shifting* and *padding*, which create a framework for contextually linking the past with the predicted future, while avoiding any accumulation of prediction errors. Extensive experiments on three well-known datasets revealed that our strategies when applied to RNN and CNN backbones, outperform existing methods. Our experiments also suggest a relationship between the amount of shifting and padding and the periodicity of the time series.

## 1 Introduction

Conventional time series forecasting is widely used to predict a set of target variables at a future time point based on past data collected over a predetermined length. *Next-step* forecasting (Montgomery et al., 2015; Shi et al., 2022) refers to predicting the target variables at a time point one step into the future where the unit of time is the time granularity of the measurements. *Multi-horizon* forecasting (Quaedvlieg, 2021) predicts the target variables multiple steps into the future Capistrán et al. (2010). Accurate forecasting allows people to do better resource management and optimization decisions for critical processes (Cinar et al., 2017; Salinas et al., 2020; Rangapuram et al., 2018). Applications include probabilistic demand forecasting in retail (Böse et al., 2017), dynamic assignments of beds to patients (Zhang & Nawata, 2018), monthly inflation forecasting, and much more.

Good multi-horizon forecasting requires historical data of the target variables from which to learn long-term patterns. In addition, it also requires measurements from heterogeneous data sources of useful *covariates*, often from the recent past. However, in many applications, some of the covariates can also be predicted with reasonable accuracy for the immediate future. We refer to such covariates as *future covariates*. For example, in some applications, a covariate of interest could be "precipitation", for which it is possible to use historical data as well as reasonably accurate predictions for the near future, which may be obtained from the weather service. Despite its importance, only limited approaches exist that use future covariates to improve time series predictions. Related methods can be mainly categorized into direct strategy using sequence-to-sequence models (Mariet & Kuznetsov, 2019) and iterated methods using autoregressive models (Sahoo et al., 2020).

Traditional methods to incorporate future covariates have major limitations. We propose two novel feature representation strategies to solve those limitations – *shifting* and *padding*, which create a framework for contextually linking the past with the predicted future, while avoiding any accumulation of prediction errors. Extensive experiments on three well-known datasets revealed that our strategies when applied to RNN and CNN backbones, outperform existing methods.

**Iterative methods.** The iterative strategy recursively uses a Next-step model multiple times where the predicted values for the previous time step is used as the input to forecast the next time step, as in Salinas et al. (2020). For the prediction at time step $t$, the target values $z_{t-1}$ at the previous time step, the (predicted) covariates $x_t$ for the current time step, and the context vectors $h_{t-1}$ that summarize the representation information of all the past time steps are considered as the input to predict target values $z_t$ at the current time step using RNNs. Rangapuram et al. (2018) adopted a similar approach by parameterizing a per-time-series linear state space model with recurrent neural networks. Related work with the iterative approach is in Li et al. (2019) where the basic architecture used was the transformer model with convolutional layers.

**Direct methods.** The direct method typically uses an *encoder* model to learn the feature representation of past data, which is saved as context vectors in a hidden state. A *decoder* model is utilized to intake future covariates and context vectors from the encoder and to then predict the outputs for multi-horizon forecasting, as shown in Figure 1. The multi-horizon Quantile Recurrent Forecaster by Wen et al. (2017) used an LSTM as the encoder to generate context vectors, which are combined with predicted future covariates and fed into a multi-layer perceptron (MLP) to predict the future horizon. Some works ((Fan et al., 2019; Du et al., 2020)) have applied a temporal attention mechanism between the encoder and the decoder. This architecture is able to learn the relevance of different parts of the feature representations from historical data by computing attentional weights. The weighted feature representations are then passed into the decoder to make predictions for future time steps. In Fan et al. (2019), bi-directional LSTMs are used as the decoder backbone allowing past and predicted features to be considered at every future time step. Temporal Fusion Transformer (Lim et al. (2021)) combined gated residual networks (GRNs) and an *attention mechanism* (Vaswani et al., 2017) as an additional decoder on top of the traditional encoder-decoder model. They used GRNs to filter unnecessary information and the additional decoder with attention mechanism to capture long-term dependencies between the time steps.

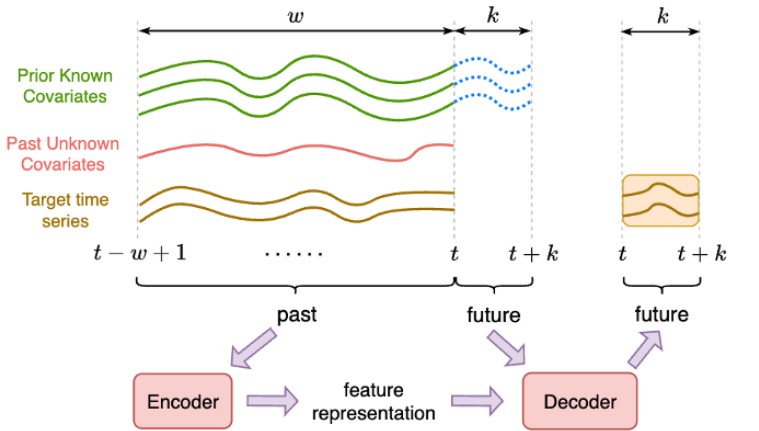

Figure 1: Direct method using sequence-to-sequence models.

The iterative as well as the direct methods aim to incorporate future covariates as inputs but suffer from several shortcomings. The iterative methods accumulate prediction errors because the input to each time step is the output from the previous step, causing the model performance to quickly degrade for longer forecasting horizons. On the other hand, direct methods are prone to miss some interactions between data from past and future time points. The encoder processes only past data, while the decoder merely concatenates past data and future covariates, which may miss specific relationships between the past and future time points. In this paper, we aim to resolve the shortcomings of both the approaches with a novel architecture.

**Our contributions:**   In this paper, we present two novel strategies to combine historical data and predicted future covariates in a meaningful manner using the strategies of *shifting* and *padding*. The architecture facilitates forecasting by learning useful feature representations from both the past and the predicted future. These two strategies transform the dataset and construct training pairs (input features, labels) used for learning. Their input features are composed of both past information on all features, predicted future information on some covariates (with appropriate time label) and a set of covariates for which no accurately predicted future information is available. The goal is to directly predict the target variables for the given horizons. The two strategies are briefly described below.

- **Shifting**: Predicted future covariates are shifted past in time and paired with appropriate past time points to create a modified input vector for the predictions. The shift length is a hyperparameter of the strategy.

- **Padding**: Covariates that cannot be accurately predicted are simply copied over from the recent past, and provided as additional input parameters after combining them with future predicted covariates. The length of the padding is a hyperparameter of the strategy.

## 2   NOTATION

Let $\mathbb{Z}_{1:t}^N = (z_{1:t}^n)_{n=1}^N = (z_1^n, z_2^n, ..., z_t^n)_{n=1}^N$ be $N$ univariate time series of target variables, where $z_t^n \in \mathbb{R}$ denotes the value of the $n$-th target variable at time $t$. Similarly, let $\mathbb{X}_{1:t}^M = (x_{1:t}^m)_{m=1}^M = (x_1^m, x_2^m, ..., x_t^m)_{m=1}^M$ be $M$ time-varying observed covariates measured until time $t$, and $\mathbb{Y}_{1:l}^Q = (y_{1:t}^q)_{q=1}^Q = (y_1^q, y_2^q, ..., y_t^q)_{q=1}^Q$ be $Q$ time series covariates that can be accurately predicted in the near future. To predict target time series at future multiple horizons from $t + 1$ to $t + k$, we define forecasting models that use the past data with a fixed length of $w$, i.e., for time $t - w + 1 : t$. The goal of such a forecasting model is to predict the trajectory $\mathbb{Z}_{t+1:t+k}^N$ of the target variables at the next $k$ time points using the past $w$ time points of all time series (targets and covariates) and $k$ future time points of future predicted covariates. The model is mathematically expressed as follows:

$$\mathbb{Z}_{t+1:t+k}^N = \mathbb{F}(\mathbb{Z}_{t-w+1:t}^N, \mathbb{X}_{t-w+1:t}^M, \mathbb{Y}_{t-w+1:t+k}^Q; \Theta),\tag{1}$$

where $\mathbb{F}(\cdot)$ is a function, $\Theta$ denotes the learnable parameters, $w$ represents the length of the history used, and $k$ is the length of forecasting horizon.

## 3   METHODOLOGY

In this section, we provide details on the learning model mentioned in Eq. (1) above. We present two different strategies, *shifting* and *padding*, to achieve this.

### 3.1   THE *Shifting* STRATEGY

In the *shifting* strategy, we focus on exploiting the predicted covariates for the future time period of interest. This is achieved by shifting the predictions of the future covariates back in time by $s$ time steps, such that the present covariates are aligned and fused with a predicted covariate $s$ time points into the future to produce distinct feature vectors at each time point (Fig. 2). Thus, the input features are composed of all time series (target and covariates) aligned from time points $t - w + 1$ to $t$ with future covariates from time points $t - w + 1 + s$ to $t + s$. Target variables are predicted for the forecasting horizon from $t + 1$ to $t + k$. This explicit use of the the predicted future covariates differentiates this approach from traditional forecasting appraoches and is expected to improve the deep learning models since it simultaneously learns from the past and the future. The predicted future covariates are shown as a blue dashed trajectory in Fig. 2. The target values computed at time $t + 1$ and later using our shifting approach are expressed as follows:

$$\mathbb{Z}_{t+1:t+k}^N = \mathbb{G}(\mathbb{Z}_{t-w+1:t}^N, \mathbb{X}_{t-w+1:t}^M, \mathbb{Y}_{t-w+1:t}^Q, \mathbb{Y}_{t-w+1+s:t+s}^Q; \Theta),\tag{2}$$

where $\mathbb{G}(\cdot)$ is the model function, $\mathbb{Z}_{t+1:t+k}^N$ represents the $N$ target variables to be predicted at the future time points $t+1 : t+k$ (region shaded brown in Fig. 2), $\mathbb{Z}_{t-w+1:t}^N$ represent the target variables from the past $w$ time points (brown trajectories in Fig. 2), $\mathbb{X}_{t-w+1:t}^Q$ and $\mathbb{Y}_{t-w+1:t}^Q$ represent the

covariates from the past $w$ time points (green and red trajectories from Fig. 2), and $\mathbb{Y}^{Q}_{t-w+1+s:t+s}$ represent the predictable future covariates along with predictions from $s$ time points into the future and then shifted back by $s$ time steps (green trajectories merged with dashed blue trajectories in Fig. 2), and $\Theta$ is the set of learnable parameters. The shift amount $s$ is a hyperparameter of the model.

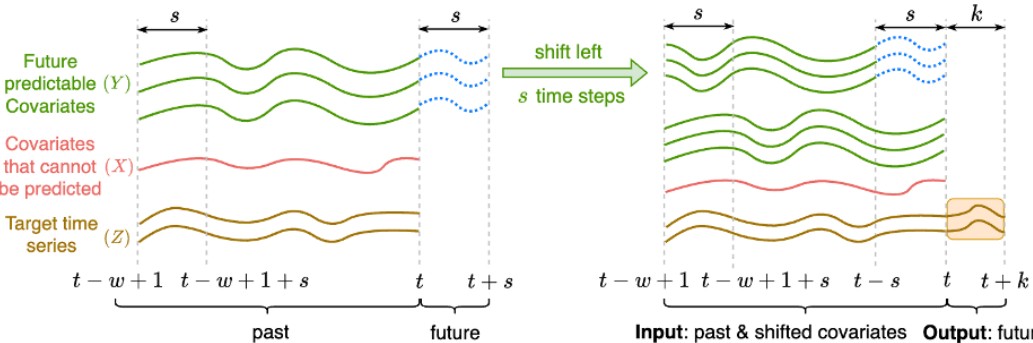

Figure 2: Input data transformed by shifting the predicted future covariates. Left: Original trajectories of all variables. Right: All input features with the shifted predicted future covariates included.

## 3.2 THE *Padding* STRATEGY

The *padding* approach attempts to extrapolate and incorporate future covariates that cannot be accurately predicted by simply making a copy of the values from the previous $s$ time points. The padded values are then combined with the values of future covariates that can be accurately predicted. More formally, the padding method makes a copy of $\mathbb{X}^{M}_{t-s:t}$ and $\mathbb{Z}^{N}_{t-s:t}$, and makes them the padded values (Fig. 3). Such manipulations can be viewed as creating a future pseudo-time $\tilde{T} = (t+1, t+s)$ where target variables $\tilde{\mathbb{Z}}^{N}_{t+1:t+s}$ (brown dashed line in Fig. 3) and covariates $\tilde{\mathbb{X}}^{M}_{t+1:t+s}$ (red dashed line in Fig. 3) repeat the previous pattern from $t - s$ to $t$. For predictable future covariates, the padding is achieved with the best predictions instead of the copies from the recent past. Eq. (3) provides a mathematical description of the padding forecasting model:

$$\mathbb{Z}^{N}_{t+1:t+k} = \mathbb{H}(\tilde{\mathbb{Z}}^{N}_{t-w+1:t+s}, \tilde{\mathbb{X}}^{M}_{t-w+1:t+s}, \mathbb{Y}^{Q}_{t-w+1:t+s}; \Theta), \tag{3}$$

where $\mathbb{H}(\cdot)$ is a model function, $\mathbb{Z}^{N}_{t+1:t+k}$ represents the $N$ target variables to be predicted at the future time points $t+1 : t+k$ (region shaded brown in Fig. 3), $\tilde{\mathbb{Z}}^{N}_{t-w+1:t+s} = \mathbb{Z}^{N}_{t-w+1:t} + \mathbb{Z}^{N}_{t+1-s:t}$ is the concatenation of the time series in the range $t - w + 1 : t$ with a copy of the time series in the range $t + 1 - s : t$ (brown trajectories in Fig. 3), $\tilde{\mathbb{X}}^{M}_{t-w+1:t+s} = \mathbb{X}^{M}_{t-w+1:t} + \mathbb{X}^{M}_{t+1-s:t}$ is the set of padded covariates corresponding to covariates that cannot be accurately predicted (pink trajectories in Fig. 3), $\mathbb{Y}^{Q}_{t-w+1:t+s}$ is the set of predicted future covariates padded with the predictions for the time range $t + 1 : t + s$ (green trajectory in Fig. 3), and $\Theta$ is the set of learnable parameters.

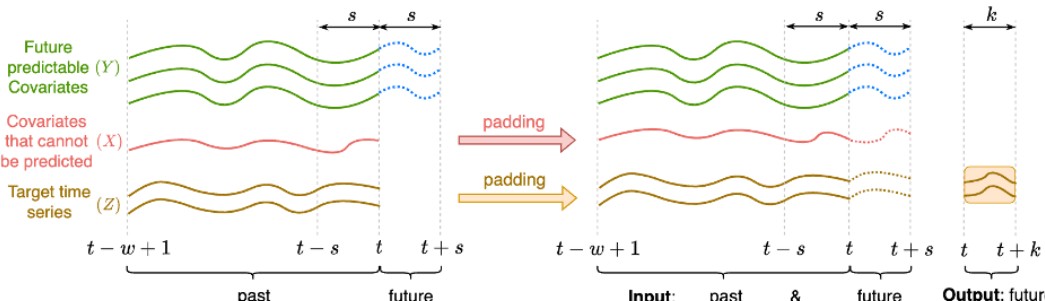

Figure 3: Input data transformed using the padding strategy. Pink and brown dotted trajectories represent the padded information of the future covariates that cannot be accurately predicted and the target variables being modeled, while the blue dotted trajectories represent the predicted future covariates. Left: Original trajectories of all variables with the predicted future covariates. Right: All input features with the padded values and predicted future covariates and the output trajectory.

### 3.3 NETWORK ARCHITECTURES

To validate the effectiveness of the *shifting* and *padding* strategies for feature representation, we constructed simple RNN and CNN architectures that included as input the covariates as described above in Sections 3.1 and 3.2. The networks were set to forecast $k$ future steps in one shot instead of the traditional sequential prediction to avoid accumulation of errors.

#### 3.3.1 RNN MODELS WITH SHIFTING

Shifting is implemented by providing the RNN backbone with $w$ hidden states as shown in Fig. 4. The standard RNNs was further modified to remove the hidden states $h_{t+1}, \ldots, h_{t+k}$ to enable a one-shot prediction of the $k$ future time points. The input to each hidden state $h_j$ were variables associated with time $t = j$ (i.e., target variable $z_j$ and covariates $x_j, y_j$), as well as the predicted covariates (shifted by $s$), $y_{j+s}$ as shown in Fig. 4. RNN model generates the output for a target variables $(z_{t+1}, \ldots, z_{t+k})$ in a one-shot manner. The hidden states are described as follows:

$$h_j = f(h_{j-1}, z_j, x_j, y_j, y_{j+s}),  \tag{4}$$

where $f$ is an activation function; $h_j$ and $h_{j-1}$ refer to the current and previous hidden states; $z_j, x_j$, and $y_j$ represent the target time series, future covariates that cannot be accurately predicted, and predictable future covariates from the past $w$ steps; and $y_{j+s}$ denotes the predicted future covariates from $k$ steps into the future.

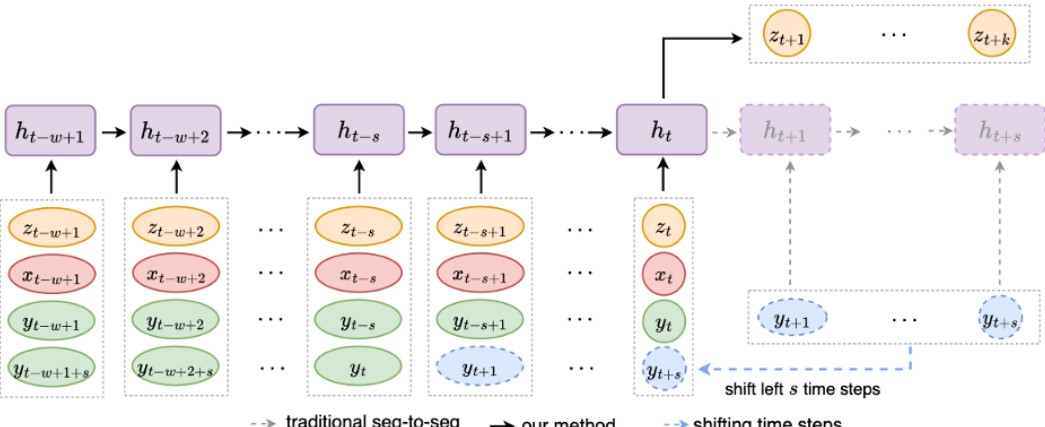

Figure 4: RNN models with the shifting strategy. Dashed ovals represents predicted future covariates that have been shifted. Solid ovals are historical data or data for which we do not have future predictions available. Colors are same as before.

#### 3.3.2 RNN MODELS WITH PADDING

The RNN backbone for padding is similar to that for shifting, but extended with $s$ extra states for pseudo-times, as shown in Fig. 5. A precise mathematical formulation is given below.

$$h_j = \begin{cases} f(h_{j-1}, z_j, x_j, y_j), & j \in [t-w+1, t] \\ f(h_{j-1}, \tilde{z}_j, \tilde{x}_j, \tilde{y}_j), & j \in [t+1, t+s], \end{cases}  \tag{5}$$

where $f(\cdot)$ is the activation function; $h_j$ and $h_{j-1}$ refer to the current and previous hidden states; $z_j, x_j, y_j$ represent the target variables, covariates that cannot be accurately predicted, and covariates that can be accurately predicted, all for the past time points $j \in [t-w+1, t]$; and $\tilde{z}_j, \tilde{x}_j, \tilde{y}_j$ represent the padded versions of the same variables for the future time range.

#### 3.3.3 CNN MODELS WITH SHIFTING

Convolutional Neural Networks (CNNs) can summarize and learn from the input data using sliding filters that extract features with convolutional computations. The time series are aligned in a manner similar to how we handled RNNs with shifting. Each CNN filter is used to extract features in parallel by sliding from the first time point to the last time point, as shown on the left in Fig. 6. The parallel

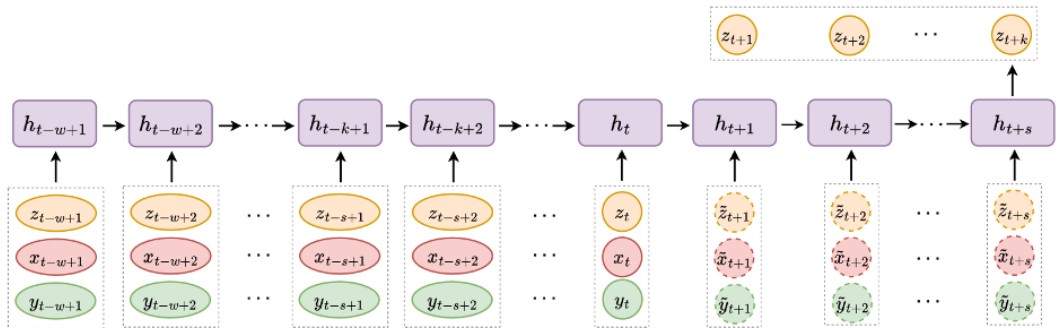

Figure 5: RNN models with the padding strategy. Dashed ovals represents padded features. Solid ovals before time $t$ are historical data. Colors are same as before.

action of the filters allows for the simultaneous learning of past and (predicted) future information. As with RNN, CNN models will output predictions of the target variables for the desired range of time points (i.e., horizon) in one shot.

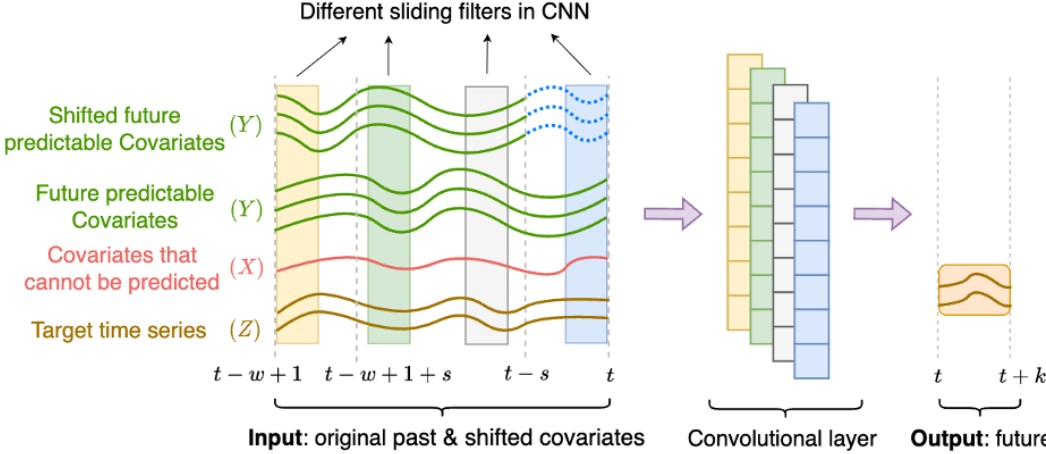

Figure 6: CNN models with the shifting strategy. Left: Input from original past and shifted future. Right: Actual future target time series. Blue line represents prior known future covariates. Pink, brown lines are past unknown future covariates and target time series, respectively.

#### 3.3.4 CNN MODELS WITH PADDING

The time series are aligned similar to how it was organized for RNNs with padding. The sliding filters are similar to that used in CNNs with shifting, but with $t + s$ time points, as shown in Fig. 7. CNNs with padding require more scanning because of the extra padded pseudo-times ($t + 1 : t + s$), but the actual filters are more compact because fewer time series are processed.

## 4 EXPERIMENTS

### 4.1 DATASETS

Three real world datasets were used for time series forecasting tasks in this paper. *Beijing PM2.5* and *Electricity price* datasets are publicly available from UCI and Kaggle repositories, respectively. The third one is the *Water stage* dataset downloaded from the South Florida Water Management District website. More details about dataset descriptions can be found in Appendix A.

**Beijing PM2.5** It includes hourly observed data from January 1, 2010, to December 31, 2014. We consider PM2.5 as the target variable to predict, other variables such as dew, temperature,

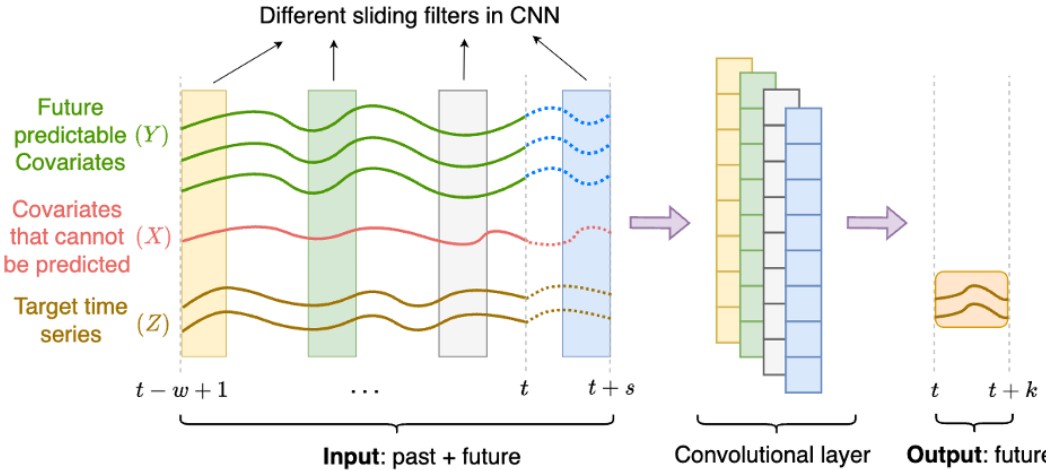

Figure 7: CNN models with the padding strategy. Left shows inputs from the past with padded series. Filters create convolutional layers, which are then used to output the target variables on the right. Colors are used as in previous figures.

pressure, wind speed, wind direction, snow, and rain are covariates that can be predicted and can influence PM2.5 values. $PM2.5 \in [0, 671]\ \mu g/m^3$ in this dataset.

**Electricity price** It was recorded every hour from January 1, 2015, to December 31, 2018. `Energy_dataset.csv` includes energy demand, generation, prices, while `weather_features.csv` contains features such as temperature and humidity. Electricity price is the target variable to predict, while prior known covariates are energy demand, generation, and weather features. *Electricity price* $\in$ [\$9.33, \$116.8] in this dataset[1].

**Water stage** This is an hourly dataset from January 1, 2010, to December 31, 2020 and includes information on water levels, the height of gate opening, water flow values through the gate, water volumes pumped at gates, and rainfall measures. Water stage is the target variable while other variables are covariates. Rainfall, gate position, and pump control are future covariates that can be predicted. *Water stage* $\in [-1.25, 4.05]$ feet in this dataset.

### 4.2 TRAINING AND EVALUATION

For each of datasets, we used the first 80% as a training set to train the model and the last 20% as a test set to evaluate the performance. During the training phase, common techniques such as normalization, dropout, regularization were used to avoid overfitting. Grid search was used to fine-tune the models for optimal hyperparameters including the number of layers, the number of neurons, learning rate, batch size, regularization factor, and the number of epochs. Mean Absolute Error (MAE) and Root Square Mean Error (RSME) were used to evaluate models. Details on the training and evaluation process is in Appendix B.

The four deep learning models (with RNNs or CNNs, and with shifting or padding) were tested with the three datasets. Seq-to-Seq models (Du et al., 2020), MQRNN (Rangapuram et al., 2018), DeepAR (Salinas et al., 2020), Temporal Fusion Transformer (TFT) (Lim et al., 2021) were compared to the four methods mentioned proposed in this manuscript. We used horizon values of $k = 6, 12, 24$, and 48 hours to forecast with input windows of size $w = 72$ hours and predictable future covariates from $s$ time steps ahead.

### 4.3 RESULTS

The results are summarized in Tables 1, 2, and 3. The lowest errors in each column are in bold font. The results show that the *shifting* and *padding* methods with CNNs outperform all the other

---

[1]Although this has no bearing on the analysis or conclusions, the currency for the price column in this dataset was unavialable. The \$ sign is used as a proxy for whatever currency was intended.

Table 1: MAE & RMSE for the *Beijing PM2.5* dataset. (PM2.5 values $\in [0, 671] \mu g/m^3$)

|  | Methods | $k = 6$ hrs | | $k = 12$ hrs | | $k = 24$ hrs | |
| --- | --- | --- | --- | --- | --- | --- | --- |
|  |  | MAE | RMSE | MAE | RMSE | MAE | RMSE |
| Baselines | Seq-to-Seq | 25.57 | 41.66 | 30.64 | 49.01 | 35.61 | 54.98 |
|  | MQRNN | 39.95 | 55.66 | 46.56 | 60.22 | 36.63 | 52.02 |
|  | DeepAR | 37.53 | 57.24 | 42.04 | 65.16 | 49.02 | 72.88 |
|  | TFT | 33.15 | 55.65 | 33.84 | 53.59 | 37.69 | 59.17 |
| Our methods | Shifting (RNN) | 24.80 | 40.62 | 30.47 | 47.74 | 36.81 | 55.52 |
|  | Padding (RNN) | 24.73 | 40.00 | 29.26 | 46.89 | 36.67 | 55.03 |
|  | Shifting (CNN) | **23.79** | **38.81** | **28.49** | **45.58** | **33.51** | 51.61 |
|  | Padding (CNN) | 24.29 | 39.26 | 29.30 | 45.87 | 34.11 | **51.14** |

Table 2: MAE & RMSE for *Electricity price* dataset. (Price $\in [\$9.33, \$116.8]$)

|  | Methods | $k = 6$ hrs | | $k = 12$ hrs | | $k = 24$ hrs | |
| --- | --- | --- | --- | --- | --- | --- | --- |
|  |  | MAE | RMSE | MAE | RMSE | MAE | RMSE |
| Baselines | Seq-to-Seq | 2.769 | 3.761 | 3.350 | 4.596 | 3.736 | 5.169 |
|  | MQRNN | 3.750 | 4.840 | 7.019 | 7.905 | 8.33 | 9.609 |
|  | DeepAR | 5.444 | 7.201 | 6.483 | 8.823 | 7.242 | 9.993 |
|  | TFT | 5.953 | 8.849 | 6.618 | 8.779 | 7.449 | 9.986 |
| Our methods | Shifting (RNN) | 2.938 | 4.042 | 3.500 | 4.764 | 3.721 | 4.876 |
|  | Padding (RNN) | **2.437** | **3.454** | **3.053** | **4.167** | **3.541** | **4.704** |
|  | Shifting (CNN) | 2.681 | 3.577 | 3.232 | 4.349 | 3.667 | 4.969 |
|  | Padding (CNN) | 2.690 | 3.710 | 3.226 | 4.359 | 3.583 | 4.745 |

Table 3: MAE & RMSE for the *Water Stage* dataset. (Water Levels $\in [-1.25, 4.05]$ ft)

|  | Methods | $k = 12$ hrs | | $k = 24$ hrs | | $k = 48$ hrs | |
| --- | --- | --- | --- | --- | --- | --- | --- |
|  |  | MAE | RMSE | MAE | RMSE | MAE | RMSE |
| Baselines | Seq-to-Sequence | 0.123 | 0.162 | 0.134 | 0.174 | 0.136 | 0.176 |
|  | MQRNN | 0.123 | 0.149 | 0.123 | 0.225 | 0.126 | 0.176 |
|  | DeepAR | 0.133 | 0.194 | 0.146 | 0.231 | 0.127 | 0.195 |
|  | TFT | 0.057 | 0.083 | 0.089 | 0.139 | **0.077** | 0.109 |
| Our methods | Shifting (RNN) | 0.119 | 0.167 | 0.132 | 0.171 | 0.158 | 0.207 |
|  | Padding (RNN) | 0.113 | 0.154 | 0.133 | 0.181 | 0.134 | 0.177 |
|  | Shifting (CNN) | 0.074 | 0.097 | 0.080 | 0.105 | 0.087 | 0.115 |
|  | Padding (CNN) | **0.053** | **0.076** | **0.068** | **0.092** | 0.080 | **0.105** |

methods compared in this paper for two of the datasets. The methods with RNNs are sensitive to the forecasting length, $k$. They outperform the "Baseline" methods for short-term predictions ($k = 6, 12$ hrs), but often fail to do so for long-term forecasting ($k = 24, 48$ hrs). Our experiments suggest that the future covariates play a significant role in time series predictions. It is also surprising that the simple models (RNNs and CNNs) outperform the sophisticated and complicated deep neural networks (labeled "Baseline" in the Tables).

We also experimented with the hyperparameter $s$, which refers to extent of shifting or padding. As expected, the performance is sensitive to the choice of $s$. The best performance appears to be achieved when $s = k$. In many applications, we may be able to reliably predict future covariates for a window much larger than the $k$ used here. However, when $s > k$, either the performance is flat or deteriorates as $s$ is increased. When $s > w$, the performance appears to deteriorate rapidly. Our experiments also allowed us to consider if $s$ is impacted by the periodicity of the datasets.

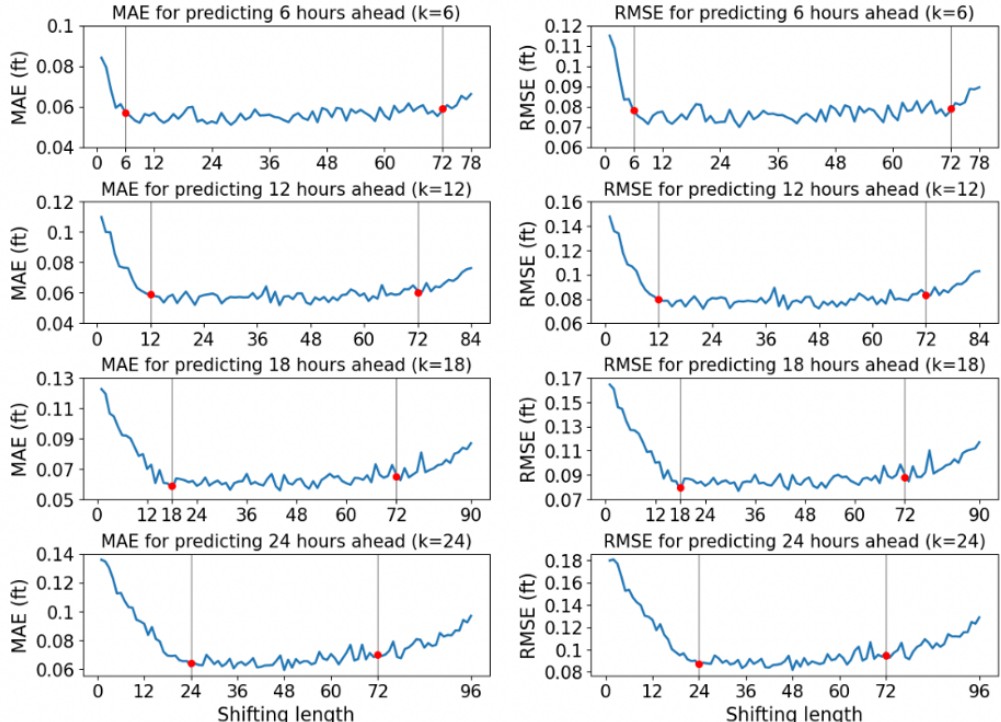

Figure 8: MAE & RMSE for different forecasting lengths ($k$) and shift lengths ($s$). Left red point of each subplot represents the errors when $s = k$ while right red point denotes the errors when $s = w$.

The graphs in Fig. 8 shows the MAE and RMSE values as a function of shift length, $s$, for different values of $k$, making sure to consider values of $s$ that range from smaller than $k$ (prediction horizon) all the way to larger than $w$ (input window size).

## 5    DISCUSSION AND CONCLUSIONS

Simple modifications to the basic RNN and CNN architectures have allowed us to build deep learning models that outperform more sophisticated models. Our experiments suggest that the utilization of future covariates (whether predictable or not) can enhance performance considerably. Furthermore, our experiments have helped us to delineate the relationship between the shifting and padding lengths and the model performance. We have observed that $s = k$ results in the best performance since future covariates in exact same forecasting horizon included. If $s < k$, we only get to utilize some of the predicted covariates from the future for the prediction horizon of $k$ time steps resulting in considerably lower performances. If $s > w$, then we end up dropping some of the future covariates in order to align with the input window, which again results in considerably lower performances.

We also observed that there is considerable periodicity and seasonality in the datasets we used in our experiments. However, in the range $k <= s <= w$, the variations in performance was too small to be significant. While there were some local minima in the performance when $s$ was a multiple of the period $p$, the improvements were not significant.

In conclusion, the *shifting-* and *padding*-based CNNs proposed in this paper outperformed all the baseline deep learning methods considered for our experiments. The corresponding methods with RNNs outperformed the baseline methods for relatively short-term prediction ($k = 6, 12$ hrs), but was not as accurate for longer term forecasting ($k = 24, 48$ hrs). The critical feature of our methods appears to be the use of future covariates, either by shifting or padding. *Shifting*-based CNNs performed best when the shift length $s$ is between the forecast horizon and the length of input window ($k \leq s \leq w$), suggesting a sweet spot for how much of the future is needed for good performance. Periodicity in the datasets appears to have small, but insignificant influence on the performance of the *shift*-based methods.

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

## A    APPENDIX: DATASET

### A.1    BEIJING PM2.5 DATASET

This dataset is Beijing Air Quality data set from the public UCI Website `https://archive.ics.uci.edu/ml/datasets/Beijing+PM2.5+Data`. It includes hourly observed data from January 1, 2010, to December 31, 2014. The data set has 43,824 rows and 13 columns. The first column is simply an index and was ignored for the analysis. The four columns labeled as year, month, day, and hour, were combined into a single feature called "year-month-day-hour". We consider PM2.5 as the target variable to predict, other variables such as dew, temperature, pressure, wind speed, wind direction, snow, rain, etc. are the prior known covariates that can influence PM2.5 values. $PM2.5 \in [0, 671]\ \mu g/m^3$ in this dataset.

Table 4: Description of Beijing PM2.5 Dataset

| VARIABLE | DESCRIPTION |
|---|---|
| No | row number |
| year | year of data in this row |
| month | month of data in this row |
| day | day of data in this row |
| hour | hour of data in this row |
| pm2.5 | PM2.5 concentration $(ug/m^3)$ |
| DEWP | Dew Point $(\hat{a},, f)$ |
| TEMP | Temperature $(\hat{a},, f)$ |
| PRES | Pressure $(hPa)$ |
| cbwd | Combined wind direction |
| Iws | Cumulated wind speed $(m/s)$ |
| Is | Cumulated hours of snow |
| Ir | Cumulated hours of rain |

### A.2    ENERGY (ELECTRICITY) PRICE DATASET

This dataset contains 4 years of electrical consumption, generation, pricing, and weather data for Spain. It is publicly available in the Kaggle website: `https://www.kaggle.com/datasets/?search=hourly+energy+demand`. It has two hourly datasets from January 1, 2015, to December 31, 2018. Energy_dataset.csv includes the information of energy demand, generation, prices, and weather_features.csv gives the weather features temperature, humidity, etc. Electricity price is the target variable to predict, while prior known covariates are energy demand, generation, and weather features. *Electricity price* $\in [9.33, 116.8]$ in this dataset.

Table 5: Description of Energy (electricity) price dataset

| VARIABLE | DESCRIPTION |
| --- | --- |
| generation biomass | biomass generation in $MW$ |
| generation fossil brown coal/lignite | coal/lignite generation in $MW$ |
| generation fossil coal-derived gas | coal gas generation in $MW$ |
| generation fossil gas | gas generation in |
| generation fossil hard coal | coal generation in $MW$ |
| generation fossil oil | oil generation in $MW$ |
| generation fossil oil shale | shale oil generation in $MW$ |
| generation fossil peat | peat generation in $MW$ |
| generation geothermal | geothermal generation in $MW$ |
| generation hydro pumped storage aggregated | hydro1 generation in $MW$ |
| generation hydro pumped storage consumption | hydro2 generation in $MW$ |
| generation hydro run-of-river and poundage | hydro3 generation in $MW$ |
| generation hydro water reservoir | hydro4 generation in $MW$ |
| generation marine | sea generation in $MW$ |
| generation nuclear | nuclear generation in $MW$ |
| generation other | other generation in $MW$ |
| generation other renewable | other renewable generation in $MW$ |
| generation solar | solar generation in $MW$ |
| generation waste | waste generation in $MW$ |
| generation wind offshore | wind offshore generation in $MW$ |
| generation wind onshore | wind onshore generation in $MW$ |
| forecast wind onshore day ahead | forecasted onshore wind generation |
| forecast solar day ahead | forecasted solar generation |
| forecast wind onshore day ahead | forecasted offshore wind generation |
| total load forecast | forecasted electrical demand |
| total load actual | actual electrical demand |
| price day ahead | forecasted price $EUR/MWh$ |
| price actual | price in $EUR/MWh4$ |

Table 6: Description of weather feature dataset

| PART | DESCRIPTION |
|---|---|
| dt_iso | datetime index localized to CET |
| city_name | name of city |
| temp | temperature in $K$ |
| temp_min | minimum in $K$ |
| temp_max | maximum in $K$ |
| pressure | pressure in $hPa$ |
| humidity | humidity in $\%$ |
| wind_speed | wind speed in $m/s$ |
| wind_deg | wind direction |
| rain_1h | rain in last hour in $mm$ |
| rain_3h | rain last 3 hours in $mm$ |
| snow_3h | snow last 3 hours in $mm$ |
| clouds_all | cloud cover in $\%$ |
| weather_id | Code used to describe weather |
| weather_main | Short description of current weather |
| weather_description | Long description of current weather |
| weather_icon | Weather icon code for website |

## A.3 WATER STAGE PREDICTION

Water stage prediction: we downloaded a dataset in the real world including the information of water stage, the height of gate opening, the amount of water flowing through the gate, the amount of pumped and rainfall from January 1, 2010, to December 31, 2020. Water stage is the target variable while other variables are covariates. Rainfall information, gate position, and pump control are prior known covariates. *Water stage* $\in [-1.25, 4.05]$ feet in this dataset.

Table 7: Description of water stage dataset

| PART | DESCRIPTION |
|---|---|
| WS_S1 | Water Stage at Station 1 in $ft$ |
| WS_S4 | Water Stage at Station 4 in $ft$ |
| FLOW_S25A | The amount of water flowing Station 25A in $m^3/s$ |
| GATE_S25A | The height of gate opening at Station 25A $m$ |
| HWS_S25A | Head water stage at Station 25A in $ft$ |
| TWS_S25A | Tail water stage at Station 25A in $ft$ |
| FLOW_S25B | The amount of water flowing Station 25B in $m^3/s$ |
| GATE_S25B | The height of gate1 opening at Station 25B $m$ |
| GATE_S25B2 | The height of gate2 opening at Station 25B $m$ |
| HWS_S25B | Head water stage at Station 25B in $ft$ |
| TWS_S25B | Tail water stage at Station 25B in $ft$ |
| PUMP_S25B | The amount of pumped water at Station 25B in $m^3/s$ |
| FLOW_S26 | The amount of water flowing Station 26 in $m^3/s$ |
| GATE_S26_1 | The height of gate1 opening at Station 26 $m$ |
| GATE_S26_2 | The height of gate2 opening at Station 26 $m$ |
| HWS_S25B | Head water stage at Station 26 in $ft$ |
| TWS_S25B | Tail water stage at Station 26 in $ft$ |
| PUMP_S26 | The amount of pumped water at Station 26 in $m^3/s$ |
| MEAN_RAIN | Mean value of rainfall of radar rainfall in $inch$ |

# B   APPENDIX: TRAINING AND EVALUATION

For each of datasets, we used the first 80% as training set to train the model and the last 20% as test set to evaluate the performance. During the training phase, common skills such as normalization, dropout, regularization were used to avoid overfitting. Mean Absolute Error (MAE) and Root Square Mean Error (RSME) are the measurement to evaluate models.

## B.1   TRAINING DETAILS

Table 8: Training settings for *Beijing PM2.5* dataset using RNN and CNN models

|  | Lr | Decay rate | Batch size | Epoch | L1 | L2 | Early Stopping |
|---|---|---|---|---|---|---|---|
| Shifting (RNN) | 1e-4 | 0.99 | 512 | 8000 | 1e-5 | 1e-4 | patience=1000 |
| Padding (RNN) | 1e-4 | - | 512 | 1500 | - | - | - |
| Shifting (CNN) | 1e-4 | 0.90 | 512 | 8000 | 1e-4 | 1e-3 | patience=500 |
| Padding (CNN) | 1e-4 | - | 512 | 1000 | - | - | - |

Table 9: Training settings for *Energy (electricity) price* dataset using RNN and CNN models

|  | Lr | Decay rate | Batch size | Epoch | L1 | L2 | Early Stopping |
|---|---|---|---|---|---|---|---|
| Shifting (RNN) | 1e-4 | 0.99 | 512 | 8000 | 1e-5 | 1e-4 | patience=1000 |
| Padding (RNN) | 1e-4 | 0.90 | 512 | 8000 | 1e-3 | 1e-2 | patience=1000 |
| Shifting (CNN) | 1e-5 | 0.90 | 512 | 8000 | 1e-4 | 1e-3 | patience=500 |
| Padding (CNN) | 1e-5 | 0.90 | 512 | 8000 | 1e-4 | 1e-3 | patience=1000 |

Table 10: Training settings for *Water Stage* dataset using RNN and CNN models

|  | Lr | Decay rate | Batch size | Epoch | L1 | L2 | Early Stopping |
|---|---|---|---|---|---|---|---|
| Shifting (RNN) | 1e-4 | - | 512 | 3000 | - | - | - |
| Padding (RNN) | 1e-4 | - | 512 | 2000 | - | - | - |
| Shifting (CNN) | 1e-4 | - | 512 | 2000 | - | - | patience=500 |
| Padding (CNN) | 1e-4 | - | 512 | 3000 | - | - | patience=1000 |

## B.2   EVALUATION DETAILS

After models have been trained with the training data (first 80% of the entire dataset), we test the trained models with the test set (last 20% of the entire dataset). Mean Absolute Error (MAE) and Root Square Mean Error (RSME) are the measurement to evaluate models.

$$MAE = \frac{\sum_{i=1}^{N} |\hat{y}_i - y_i|}{N} \tag{6}$$

$$RMSE = \sqrt{\frac{\sum_{i=1}^{N} (\hat{y}_i - y_i)^2}{N}} \tag{7}$$

where $N$ is the number of samples in the test set, $\hat{y}_i$ is the predicted value of model, $y_i$ is the actual value in the test set.

## C APPENDIX: SHIFTING STRATEGY

For the *shifting* strategy shown in Figure 2, to predict the target variables in future $k$ time steps, we shifted the *future covariates* in the same horizon to the past by exact $k$ steps to incorporate enough feature representations of future covariates. That is because the future covariates from time $t + 1$ to $t + k$ have much more influence on those target variables in the same horizon that we want to predict. However, in the real world, we can get the future covariates with the longer time range ($¿$ k). Therefore, we also tried different shifting lengths using CNNs to explore the relationship between the shifting length, $s$, the past length, $w$, and forecasting length, $k$ and possible periodicity of dataset, $p$. Here we used water stage dataset since water stage is influenced by tide whose periodicity is roughly 12 hours. The length of past data is still 3 days (i.e., $w$ = 72 hours). The corresponding MAEs and RMSEs are listed in Tables 11 and 12 and visualized in Figure 12.

### C.1 $s < k$

If $s < k$, it indicated the shorter time range of future covariates is shifted to the past, which cause covariates from $t + s$ to $t + k$ in future $k$ time steps are missed. However, we know those future covariates from time $t + 1$ to $t + k$ have much influence on the target variables in the same horizon to predict. This missing covariates will result in the higher MAEs and RMSEs shown in the left part of each subplot where $s < k$ (Figure 12).

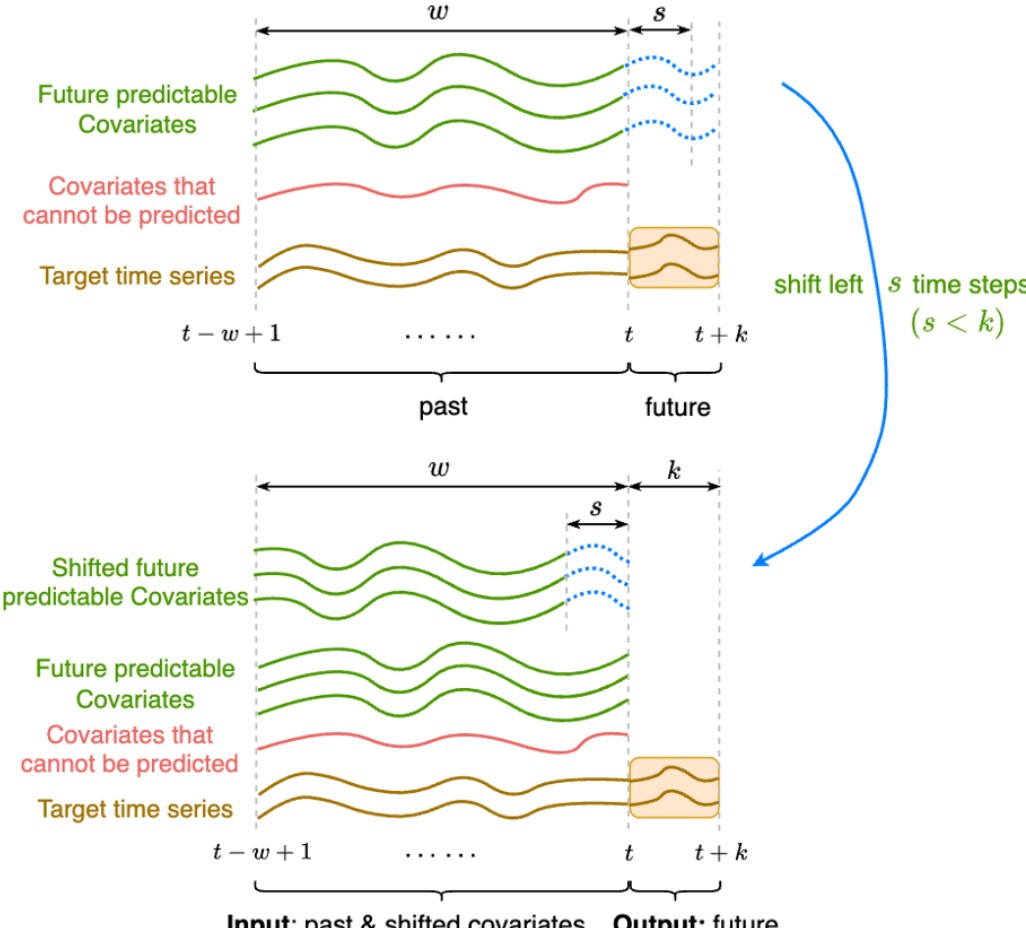

Figure 9: Input data transformed by shifting connection ($s < k$) for future covariates. Left: Original heterogeneous inputs and output. Right: Shifted heterogeneous inputs and output. Dot line represents prior known future covariates.

## C.2 $w < s <= w + k$

If $w < s < w + k$, it means a few of future covariate from time $t + 1$ to $t + s - w$ are also missed because they are shifted too many steps to the past such that exceed the window vision of past data. They would be totally out of the vision window of the past data if $s = w + k$. This missing covariates will result in the higher MAEs and RMSEs shown in the right part of each subplot where $w < s <= w + k$ (Figure 12).

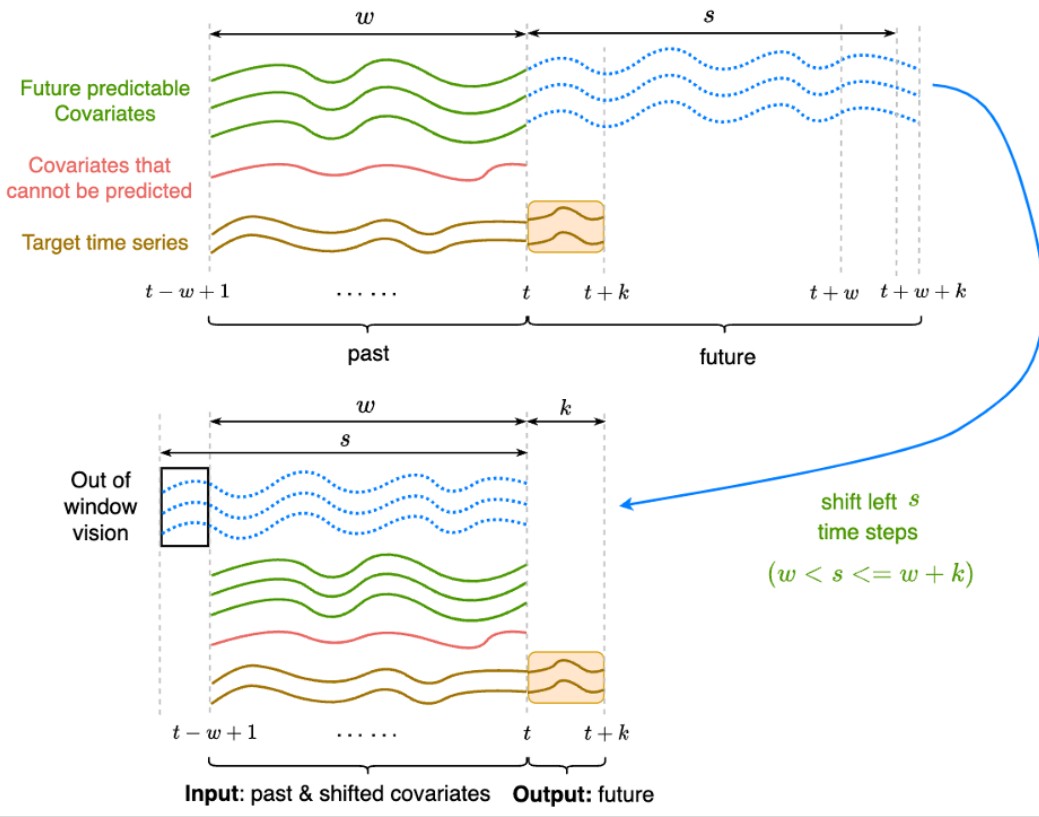

Figure 10: Input data transformed by shifting connection ($w < s <= w + k$) for future covariates. Left: Original heterogeneous inputs and output. Right: Shifted heterogeneous inputs and output. Dot line represents prior known future covariates.

## C.3 $k <= s <= w$

If the shifting length is between the forecasting length and the length of past data ($k <= s <= w$), MAE and RMSE are lower and fluctuating in a small range since all covariates from future $k$ time steps are incorporated in the window of the past data. We guess the fluctuation is caused by periodicity of the dataset since there are local minimum errors if the shifting length is approximately equal to the periodicity plus the forecasting length.

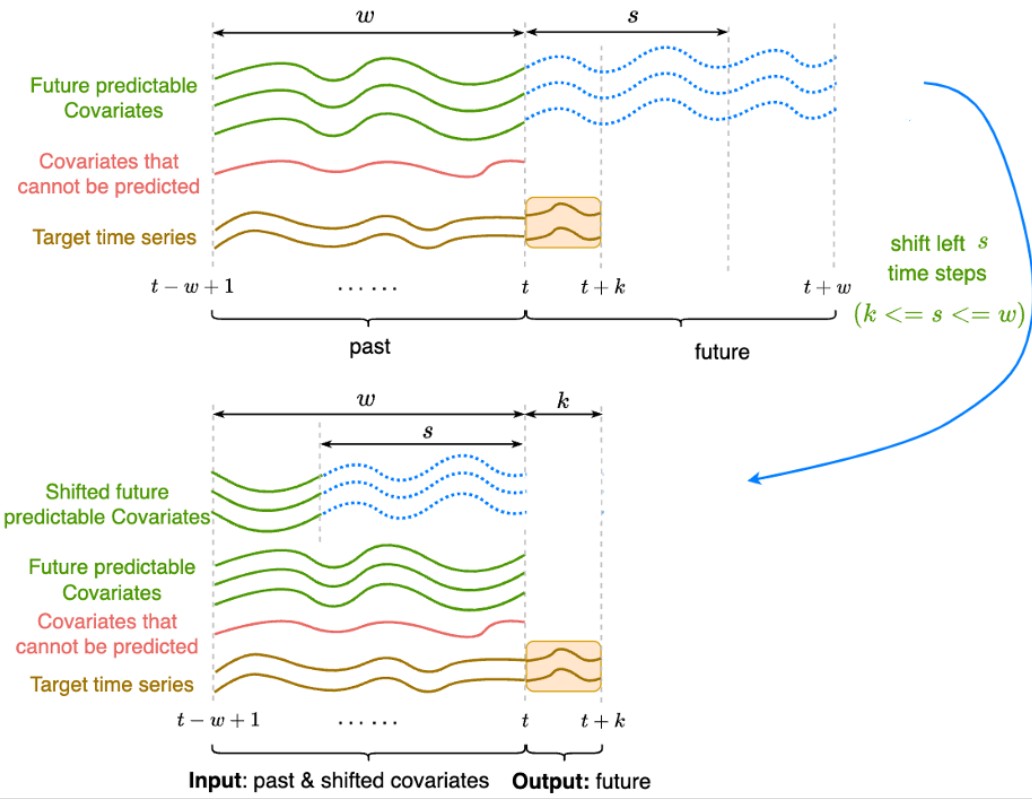

Figure 11: Input data transformed by shifting connection ($k <= s <= w$) for future covariates. Left: Original heterogeneous inputs and output. Right: Shifted heterogeneous inputs and output. Dot line represents prior known future covariates.

C.4    VISUALIZATION OF MAE & RMSE WITH DIFFERENT SHIFTING LENGTHS

With the *Water Stage* dataset, we tried different shifting lengths from 1 to $w + k$ for a certain forecasting length.

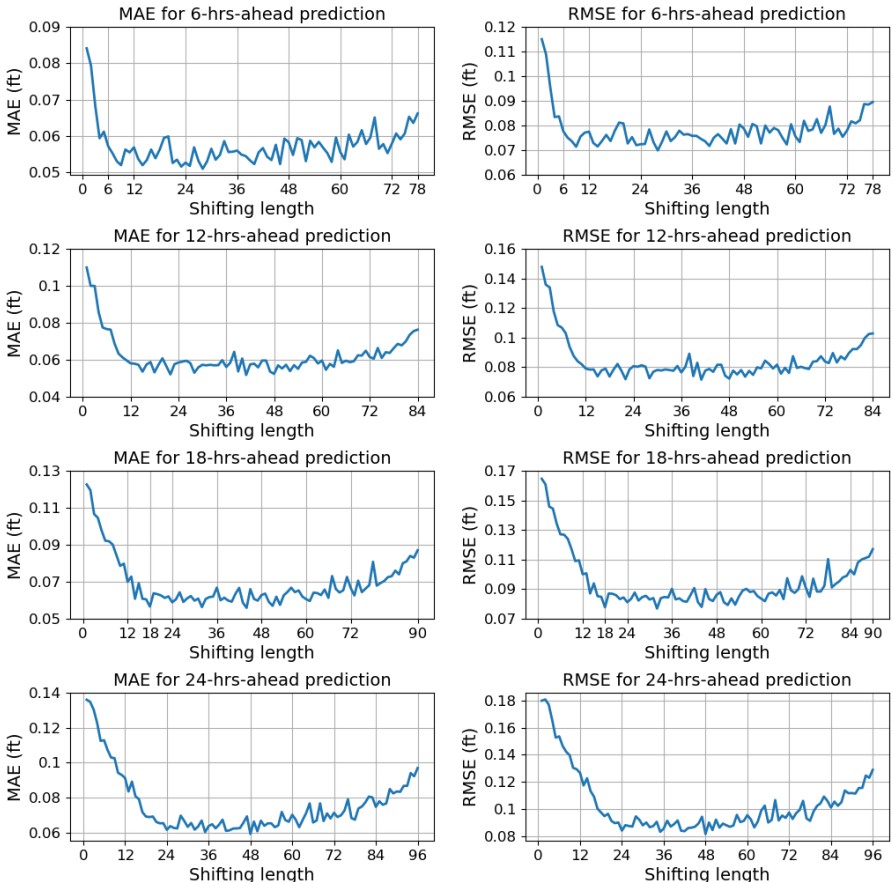

Figure 12: MAE & RMSE for different forecasting lengths ($k$) with different shifting lengths ($s$).

Table 11: MAE & RMSE for water stage dataset with different shifting lengths.

| Shifting length $s$ | $k$=6 hrs | | $k$=12 hrs | | $k$=18 hrs | | $k$=24 hrs | |
|---|---|---|---|---|---|---|---|---|
| | MAE | RMSE | MAE | RMSE | MAE | RMSE | MAE | RMSE |
| $s = 1$ | 0.0841 | 0.1150 | 0.1099 | 0.1478 | 0.1226 | 0.1646 | 0.1360 | 0.1799 |
| $s = 2$ | 0.0794 | 0.1088 | 0.1000 | 0.1358 | 0.1195 | 0.1608 | 0.1348 | 0.1808 |
| $s = 3$ | 0.0683 | 0.0952 | 0.0999 | 0.1339 | 0.1066 | 0.1458 | 0.1304 | 0.1770 |
| $s = 4$ | 0.0594 | 0.0834 | 0.0857 | 0.1177 | 0.1046 | 0.1444 | 0.1228 | 0.1656 |
| $s = 5$ | 0.0612 | 0.0837 | 0.0775 | 0.1084 | 0.0978 | 0.1344 | 0.1125 | 0.1527 |
| $s = 6$ | 0.0572 | 0.0777 | 0.0766 | 0.1068 | 0.0922 | 0.1271 | 0.1129 | 0.1536 |
| $s = 7$ | 0.0553 | 0.0751 | 0.0763 | 0.1030 | 0.0919 | 0.1268 | 0.1073 | 0.1465 |
| $s = 8$ | 0.0531 | 0.0736 | 0.0687 | 0.0936 | 0.0901 | 0.1240 | 0.1030 | 0.1423 |
| $s = 9$ | 0.0520 | 0.0714 | 0.0633 | 0.0876 | 0.0844 | 0.1171 | 0.1026 | 0.1397 |
| $s = 10$ | 0.0563 | 0.0755 | 0.0613 | 0.0840 | 0.0786 | 0.1090 | 0.0943 | 0.1304 |
| $s = 11$ | 0.0555 | 0.0771 | 0.0597 | 0.0820 | 0.0798 | 0.1095 | 0.0931 | 0.1294 |
| $s = 12$ | 0.0569 | 0.0775 | 0.0581 | 0.0791 | 0.0700 | 0.1001 | 0.0914 | 0.1266 |
| $s = 13$ | 0.0537 | 0.0729 | 0.0597 | 0.0784 | 0.0728 | 0.1007 | 0.0836 | 0.1174 |
| $s = 14$ | 0.0520 | 0.0771 | 0.0573 | 0.0785 | 0.0608 | 0.0871 | 0.0892 | 0.1227 |
| $s = 15$ | 0.0534 | 0.0737 | 0.0536 | 0.0738 | 0.0692 | 0.0939 | 0.0809 | 0.1133 |
| $s = 16$ | 0.0563 | 0.0762 | 0.0574 | 0.0779 | 0.0608 | 0.0853 | 0.0792 | 0.1091 |
| $s = 17$ | 0.0539 | 0.0737 | 0.0588 | 0.0790 | 0.0605 | 0.0848 | 0.0718 | 0.1003 |
| $s = 18$ | 0.0563 | 0.0779 | 0.0532 | 0.0737 | 0.0566 | 0.0778 | 0.0693 | 0.0975 |
| $s = 19$ | 0.0595 | 0.0812 | 0.0573 | 0.0784 | 0.0638 | 0.0871 | 0.0690 | 0.0948 |
| $s = 20$ | 0.0599 | 0.0808 | 0.0608 | 0.0822 | 0.0633 | 0.0869 | 0.0693 | 0.0965 |
| $s = 21$ | 0.0526 | 0.0727 | 0.0565 | 0.0775 | 0.0626 | 0.0859 | 0.0662 | 0.0912 |
| $s = 22$ | 0.0535 | 0.0753 | 0.0521 | 0.0719 | 0.0613 | 0.0833 | 0.0654 | 0.0896 |
| $s = 23$ | 0.0516 | 0.0720 | 0.0576 | 0.0785 | 0.0621 | 0.0844 | 0.0655 | 0.0901 |
| $s = 24$ | 0.0527 | 0.0724 | 0.0585 | 0.0807 | 0.0589 | 0.0812 | 0.0617 | 0.0841 |
| $s = 25$ | 0.0518 | 0.0725 | 0.0590 | 0.0803 | 0.0605 | 0.0834 | 0.0638 | 0.0881 |
| $s = 26$ | 0.0569 | 0.0784 | 0.0595 | 0.0813 | 0.0643 | 0.0877 | 0.0628 | 0.0875 |
| $s = 27$ | 0.0532 | 0.0732 | 0.0582 | 0.0804 | 0.0591 | 0.0823 | 0.0625 | 0.0871 |
| $s = 28$ | 0.0510 | 0.0699 | 0.0530 | 0.0725 | 0.0610 | 0.0846 | 0.0698 | 0.0946 |
| $s = 29$ | 0.0530 | 0.0735 | 0.0559 | 0.0770 | 0.0623 | 0.0855 | 0.0668 | 0.0921 |
| $s = 30$ | 0.0565 | 0.0776 | 0.0573 | 0.0781 | 0.0598 | 0.0830 | 0.0634 | 0.0880 |
| $s = 31$ | 0.0535 | 0.0737 | 0.0570 | 0.0778 | 0.0609 | 0.0840 | 0.0656 | 0.0901 |
| $s = 32$ | 0.0548 | 0.0753 | 0.0573 | 0.0785 | 0.0563 | 0.0769 | 0.0618 | 0.0866 |
| $s = 33$ | 0.0586 | 0.0779 | 0.0570 | 0.0781 | 0.0607 | 0.0839 | 0.0636 | 0.0875 |
| $s = 34$ | 0.0556 | 0.0763 | 0.0571 | 0.0776 | 0.0617 | 0.0847 | 0.0670 | 0.0906 |
| $s = 35$ | 0.0557 | 0.0765 | 0.0598 | 0.0808 | 0.0620 | 0.0844 | 0.0604 | 0.0832 |
| $s = 36$ | 0.0560 | 0.0758 | 0.0561 | 0.0765 | 0.0668 | 0.0902 | 0.0638 | 0.0856 |
| $s = 37$ | 0.0549 | 0.0758 | 0.0584 | 0.0802 | 0.0600 | 0.0831 | 0.0649 | 0.0914 |
| $s = 38$ | 0.0545 | 0.0746 | 0.0643 | 0.0891 | 0.0614 | 0.0838 | 0.0626 | 0.0871 |
| $s = 39$ | 0.0533 | 0.0736 | 0.0537 | 0.0740 | 0.0599 | 0.0819 | 0.0646 | 0.0889 |
| $s = 40$ | 0.0523 | 0.0717 | 0.0607 | 0.0830 | 0.0592 | 0.0815 | 0.0675 | 0.0916 |
| $s = 41$ | 0.0555 | 0.0750 | 0.0518 | 0.0716 | 0.0635 | 0.0860 | 0.0611 | 0.0840 |
| $s = 42$ | 0.0567 | 0.0765 | 0.0575 | 0.0777 | 0.0667 | 0.0908 | 0.0614 | 0.0837 |
| $s = 43$ | 0.0543 | 0.0747 | 0.0579 | 0.0788 | 0.0587 | 0.0812 | 0.0624 | 0.0860 |
| $s = 44$ | 0.0534 | 0.0727 | 0.0559 | 0.0769 | 0.0559 | 0.0780 | 0.0625 | 0.0863 |
| $s = 45$ | 0.0576 | 0.0785 | 0.0596 | 0.0816 | 0.0660 | 0.0900 | 0.0627 | 0.0870 |
| $s = 46$ | 0.0523 | 0.0727 | 0.0596 | 0.0817 | 0.0607 | 0.0833 | 0.0658 | 0.0894 |
| $s = 47$ | 0.0593 | 0.0804 | 0.0536 | 0.0742 | 0.0598 | 0.0820 | 0.0694 | 0.0943 |
| $s = 48$ | 0.0583 | 0.0788 | 0.0524 | 0.0723 | 0.0628 | 0.0861 | 0.0593 | 0.0816 |
| $s = 49$ | 0.0548 | 0.0754 | 0.0570 | 0.0777 | 0.0636 | 0.0880 | 0.0664 | 0.0901 |
| $s = 50$ | 0.0594 | 0.0806 | 0.0553 | 0.0751 | 0.0591 | 0.0811 | 0.0609 | 0.0845 |

Table 12: MAE & RMSE for water stage dataset with different shifting lengths.

| Shifting length $s$ | $k$=6 hrs | | $k$=12 hrs | | $k$=18 hrs | | $k$=24 hrs | |
|---|---|---|---|---|---|---|---|---|
| | MAE | RMSE | MAE | RMSE | MAE | RMSE | MAE | RMSE |
| $s = 51$ | 0.0589 | 0.0797 | 0.0573 | 0.0781 | 0.0570 | 0.0793 | 0.0669 | 0.0922 |
| $s = 52$ | 0.0531 | 0.0727 | 0.0539 | 0.0736 | 0.0619 | 0.0838 | 0.0640 | 0.0863 |
| $s = 53$ | 0.0587 | 0.0800 | 0.0571 | 0.0775 | 0.0575 | 0.0795 | 0.0654 | 0.0893 |
| $s = 54$ | 0.0569 | 0.0771 | 0.0551 | 0.0748 | 0.0627 | 0.0849 | 0.0654 | 0.0880 |
| $s = 55$ | 0.0584 | 0.0790 | 0.0585 | 0.0799 | 0.0645 | 0.0890 | 0.0632 | 0.0869 |
| $s = 56$ | 0.0568 | 0.0781 | 0.0589 | 0.0792 | 0.0668 | 0.0902 | 0.0642 | 0.0879 |
| $s = 57$ | 0.0553 | 0.0750 | 0.0622 | 0.0844 | 0.0644 | 0.0881 | 0.0719 | 0.0957 |
| $s = 58$ | 0.0529 | 0.0723 | 0.0609 | 0.0822 | 0.0653 | 0.0887 | 0.0673 | 0.0904 |
| $s = 59$ | 0.0596 | 0.0805 | 0.0580 | 0.0791 | 0.0620 | 0.0853 | 0.0663 | 0.0913 |
| $s = 60$ | 0.0556 | 0.0756 | 0.0596 | 0.0817 | 0.0606 | 0.0836 | 0.0703 | 0.0955 |
| $s = 61$ | 0.0536 | 0.0733 | 0.0546 | 0.0756 | 0.0596 | 0.0819 | 0.0675 | 0.0924 |
| $s = 62$ | 0.0604 | 0.0819 | 0.0578 | 0.0796 | 0.0641 | 0.0870 | 0.0632 | 0.0865 |
| $s = 63$ | 0.0571 | 0.0778 | 0.0562 | 0.0761 | 0.0639 | 0.0878 | 0.0673 | 0.0907 |
| $s = 64$ | 0.0584 | 0.0784 | 0.0651 | 0.0874 | 0.0628 | 0.0857 | 0.0713 | 0.0989 |
| $s = 65$ | 0.0616 | 0.0826 | 0.0583 | 0.0795 | 0.0658 | 0.0894 | 0.0768 | 0.1025 |
| $s = 66$ | 0.0578 | 0.0770 | 0.0595 | 0.0803 | 0.0613 | 0.0833 | 0.0659 | 0.0901 |
| $s = 67$ | 0.0596 | 0.0803 | 0.0587 | 0.0794 | 0.0731 | 0.0974 | 0.0670 | 0.0920 |
| $s = 68$ | 0.0607 | 0.0828 | 0.0592 | 0.0788 | 0.0661 | 0.0892 | 0.0768 | 0.1066 |
| $s = 69$ | 0.0565 | 0.0766 | 0.0624 | 0.0837 | 0.0642 | 0.0875 | 0.0667 | 0.0916 |
| $s = 70$ | 0.0578 | 0.0787 | 0.0623 | 0.0841 | 0.0654 | 0.0899 | 0.0711 | 0.0952 |
| $s = 71$ | 0.0553 | 0.0754 | 0.0649 | 0.0874 | 0.0726 | 0.0987 | 0.0682 | 0.0935 |
| $s = 72$ | 0.0579 | 0.0779 | 0.0615 | 0.0839 | 0.0666 | 0.0910 | 0.0714 | 0.0974 |
| $s = 73$ | 0.0607 | 0.0817 | 0.0606 | 0.0828 | 0.0626 | 0.0848 | 0.0690 | 0.0927 |
| $s = 74$ | 0.0591 | 0.0809 | 0.0664 | 0.0896 | 0.0706 | 0.0971 | 0.0700 | 0.0969 |
| $s = 75$ | 0.0607 | 0.0821 | 0.0610 | 0.0832 | 0.0644 | 0.0884 | 0.0727 | 0.0993 |
| $s = 76$ | 0.0653 | 0.0887 | 0.0641 | 0.0873 | 0.0663 | 0.0884 | 0.0791 | 0.1060 |
| $s = 77$ | 0.0636 | 0.0885 | 0.0637 | 0.0853 | 0.0682 | 0.0923 | 0.0684 | 0.0931 |
| $s = 78$ | 0.0662 | 0.0895 | 0.0663 | 0.0895 | 0.0808 | 0.1103 | 0.0675 | 0.0913 |
| $s = 79$ | 0.0665 | 0.0913 | 0.0686 | 0.0923 | 0.0679 | 0.0912 | 0.0739 | 0.0981 |
| $s = 80$ | 0.0679 | 0.0926 | 0.0678 | 0.0922 | 0.0693 | 0.0935 | 0.0750 | 0.1023 |
| $s = 81$ | 0.0690 | 0.0946 | 0.0699 | 0.0948 | 0.0704 | 0.0953 | 0.0774 | 0.1042 |
| $s = 82$ | 0.0717 | 0.0977 | 0.0735 | 0.0997 | 0.0725 | 0.0980 | 0.0807 | 0.1092 |
| $s = 83$ | - | - | 0.0755 | 0.1024 | 0.0730 | 0.0990 | 0.0802 | 0.1060 |
| $s = 84$ | - | - | 0.0763 | 0.1028 | 0.0760 | 0.1030 | 0.0748 | 0.1011 |
| $s = 85$ | - | - | 0.0767 | 0.1040 | 0.0740 | 0.1000 | 0.0780 | 0.1055 |
| $s = 86$ | - | - | 0.0826 | 0.1111 | 0.0800 | 0.1080 | 0.0760 | 0.1024 |
| $s = 87$ | - | - | 0.0821 | 0.1117 | 0.0810 | 0.1100 | 0.0768 | 0.1050 |
| $s = 88$ | - | - | 0.0831 | 0.1126 | 0.0840 | 0.1110 | 0.0850 | 0.1137 |
| $s = 89$ | - | - | - | - | 0.0830 | 0.1120 | 0.0825 | 0.1117 |
| $s = 90$ | - | - | - | - | 0.0870 | 0.1170 | 0.0836 | 0.1118 |
| $s = 91$ | - | - | - | - | 0.0935 | 0.1242 | 0.0834 | 0.1114 |
| $s = 92$ | - | - | - | - | 0.0896 | 0.1194 | 0.0868 | 0.1155 |
| $s = 93$ | - | - | - | - | 0.0932 | 0.1233 | 0.0868 | 0.1156 |
| $s = 94$ | - | - | - | - | 0.0925 | 0.1229 | 0.0941 | 0.1247 |
| $s = 95$ | - | - | - | - | - | - | 0.0923 | 0.1231 |
| $s = 96$ | - | - | - | - | - | - | 0.0970 | 0.1289 |
| $s = 97$ | - | - | - | - | - | - | 0.1021 | 0.1354 |
| $s = 98$ | - | - | - | - | - | - | 0.0982 | 0.1302 |
| $s = 99$ | - | - | - | - | - | - | 0.0976 | 0.1302 |
| $s = 100$ | - | - | - | - | - | - | 0.1004 | 0.1338 |

