# OpenReview forum: "NOVEL FEATURE REPRESENTATION STRATEGIES FOR TIME SERIES FORECASTING WITH PREDICTED FUTURE COVARIATES"
_ICLR.cc/2023/Conference — Submitted to ICLR 2023_

### Official Review · Reviewer_S4dZ · 2022-10-22

**Confidence:** 4
**Correctness:** 2
**Technical Novelty And Significance:** 2
**Empirical Novelty And Significance:** 2
**Recommendation:** 3

**Clarity, Quality, Novelty And Reproducibility:**

The clarity of the work is overall good, but some consistency should be improved. I list a few:
* What is “past-future connection” as in the abstract? Isn’t this connection the forecasting model?
* The predictive covariates are sometimes called predictive features (in page 2 when explaining the directive method). Similarly, the directive method is called seq-to-seq model in the end of the 2nd paragraph of Introduction.
* The sentence “... miss some interactions between data from past and future time points” is mentioned several times. What does it mean exactly? Is it the interaction between the past (past target or past covariates or both?) and future covariates? If so, it is better to make it explicit.
* “...historical data or accurate predictions…” or -> for

I don’t think the quality matches the ICLR standard, due to the concerns mentioned in the weakness. The novelty is also limited in my opinion. The reproducibility is limited, the RNN or CNN architectures are not given, some detail in the experiments are not shown (as mentioned in the weakness) and code is not available.


**Strength And Weaknesses:**

On the strength, the work studies a less explored area of how to best use the future covariates in a forecasting model, which I think is an important question.

On the weakness, I have two major concerns. The motivation of the proposed strategies is not well justified and I see some potential problems. The shifting may mess up the interaction between the covariates and target, as the time indexes are mismatched after the shifting. The padding may break the smoothness of the time series. Also, if it is a mere concatenation of past (the encoder output) and future covariates, the decoder could learn the interaction between them. For the covariates that can not be predicted well (or not observed), and if there is relevant information in them, a more principled way for me will be a multivariate method.

The other concern is lacking detail in the baselines and experiment set up. What is the baseline strategy? Some results for concatenating the covariates with the output of the encoder? Some of the baselines can also use covariates, such as DeepAR. In the experiments, did the authors try to use the covariates, at least the ones that can be predicted accurately, as input for DeepAR? Have the baselines been trained properly? Also, how did the authors generate future covariates? Do you take them from the data or do you actually predict the future covariates? In the end, these deep learning models have quite some variance in the training and some replicates are needed to have robust results. In the end, how to show the effect of the strategy and separating the influence of the model choices?


**Summary Of The Paper:**

This work proposed two strategies to use future covariates by 1) simply shifting back the covariates with prediction by length $s$ (which is a hyperparameter) and take the shifted covariates as additional input to the model; or 2) for the target and unpredicted covariates, copying the last $s$ observations to pad them into the end of original time series and then take them as input to the model. The experiments on three datasets showed better forecasting accuracy compared to 4 common deep learning baselines.

**Summary Of The Review:**

I have concerns about the motivation and the empirical results. Please refer to more detail in the weakness section.

---

### Official Review · Reviewer_nHrP · 2022-10-23

**Confidence:** 4
**Correctness:** 2
**Technical Novelty And Significance:** 1
**Empirical Novelty And Significance:** 1
**Recommendation:** 1

**Clarity, Quality, Novelty And Reproducibility:**

**Clarity.** The paper is very clearly written.

**Quality/novelty.** The technical novelty is very limited (weakness point #1). The proposed approaches are simple to state but they have not been clearly theoretically justified as to better explain when and why we should expect them to work well or not (weakness point #2). I think there are issues with the experimental results in terms of how they are presented and conducted (weakness points #3, #4, and #5). Note that even if weakness points #2 through #5 are resolved, weakness point #1 would still not be resolved in that the actual technical innovation is very limited.

**Reproducibility.** While code does not appear to be available, the proposed methods are really, really simple so I think reproducibility is not a problem. Moreover the datasets are standard.

**Strength And Weaknesses:**

Strengths:
1. The paper is well-explained.
2. The proposed methods *shifting* and *padding* are easy to understand in terms of how they are computed. The diagrams in particular are very helpful.

Weaknesses:
1. The two strategies proposed are very straightforward and do not constitute major technical advances.
2. While I understood how *shifting* and *padding* mechanically work (i.e., I understand how they are computed), I think that the authors should provide more theoretical justification for when and why we expect them to work. For example, in the case of *shifting*, it would be great if more theoretical intuition could be provided for why precisely predicted future values are helpful. Specifically, it seems like if the claim is that all the data up until the current time $t$ can very reliably predict the next $k$ future values for a few specific covariates, then doesn't it mean that the data up until the current time $t$ already contain the information that is in these particular $k$ predicted values, so actually predicting them and feeding the predicted values explicitly as inputs would somehow seem, from a theoretical standpoint, unnecessary? As a concrete example, if the next $k$ predicted values are just represented as a continuation of a line fitted to the most recent $k$ values up to time $t$, then it is unclear what the advantage is of explicitly computing the predicted $k$ values if you already have fitted a line to the most recent $k$ values. Using standard stochastic process terminology, basically I think this could be phrased in terms of the filtration up until time $t$ and that for a few covariates, the future values of these covariates are predictable processes (conditioned on the filtration up until time $t$). Using this sort of standard formalism, providing a clear argument for why shifting definitely helps would be helpful rather than only giving a seemingly heuristic justification. Separately, I think *padding* makes a much larger assumption and one can easily cook up examples where it will fail catastrophically...
3. For the training, there isn't much of any detail provided regarding how hyperparameters are tuned, which leaves me to find the results obtained to be suspicious. For example, there is no mention of a validation set (that is separate from the held out test set) that is used to help tune hyperparameters. Were hyperparameters then just tuned where one directly gets to see the test set performance (this would constitute extremely poor experimental design)? Assuming that this is not the case, then there should be a lot more details provided for how hyperparameters were selected without ever looking at the test set.
4. There are no error bars reported for Tables 1 & 2. I'd suggest adding error bars to give readers a sense of the variability of the models evaluated (for instance, you could re-run the experiment using 10 different random seeds and instead report mean/std dev of each metric across the 10 different random seeds). Without these error bars, it's hard to know just how much better one method is doing vs another. Is it just due to the random seed? If we re-run the models, is it the case that the ones currently bolded are always consistently better than the others?
5. Given how simple the proposed strategies are, I would prefer seeing a much more extensive evaluation (with many more standard publicly available datasets) so that we can better understand when they work and when they don't (i.e., when they don't provide an advantage over baseline(s)). Especially given the comments I mentioned in weakness point #2, I would be concerned that the datasets currently in the paper are too cherry-picked.

**Summary Of The Paper:**

This paper proposes two approaches to taking advantage of future values in a time series for prediction (these future values ares ones that can reliably be predicted from data up until the current time), shifting and padding. The paper shows how to use these two approaches with RNNs and CNNs and demonstrates their effectiveness on real data.

**Summary Of The Review:**

Overall, I think that this paper is not ready for publication. I encourage the authors to better justify their proposed methods from a theoretical standpoint, and to make their experimental results much more rigorous (very clearly stating how hyperparameter tuning is done so that there is no doubt that the experiments are conducted correctly, adding error bars, adding more datasets).

---

### Official Review · Reviewer_ZRck · 2022-10-23

**Confidence:** 5
**Correctness:** 4
**Technical Novelty And Significance:** 2
**Empirical Novelty And Significance:** 2
**Recommendation:** 3

**Clarity, Quality, Novelty And Reproducibility:**

No issues with clarity or quality. However, the methodological contributions are limited.

The authors did not discuss reproducibility. It is not clear whether they plan to provide the code associated with their paper.

**Strength And Weaknesses:**

Strengths
	- The paper is well-written and clear.
	- New strategies for multi-horizon forecasting are needed to avoid the accumulation of prediction errors over the forecast horizon.
	- Experimental results show better forecast accuracy compared to baselines.

Weaknesses
	- The proposed forecasting strategies are essentially heuristics, without strong mathematical foundations.
	- Given the wide variety of existing forecasting strategies, the contribution to the state-of-the-art is rather limited. Also, the methodological contributions of the paper are limited.
	- The experiments are only based on three real-world datasets. There are many other forecasting benchmarks (M5, etc) that are more relevant to the forecasting community.


Comments
- I suggest citing the paper "Forecasting: theory and practice" (https://arxiv.org/pdf/2012.03854.pdf) which discusses forecasting strategies.
- When reading the results in Tables 1-3, I am not convinced that your method "outperforms" other methods, especially seq_to_seq. The differences are rather small.  Tests of significance are needed here.


**Summary Of The Paper:**

Two novel feature forecasting strategies – shifting and padding, are proposed to avoid any accumulation of prediction errors while contextually linking the past with the predicted future. These strategies allow to better deal with predictable future covariates. For example, the problem of predicting water levels at a given location in a river or canal system using historical data and future covariates, some of which (precipitation, tide) may be predictable. Experiments performed on three datasets show the proposed strategies outperform existing methods and suggest a relationship between the amount of shifting and padding and the periodicity of the time series.


**Summary Of The Review:**

While the authors propose two new forecasting strategies with good forecast accuracy compared to baselines, the methodological contributions are limited. Furthermore, the proposed methods do not have a (strong) mathematical foundation. The experiments are limited to three (small) datasets. On the positive side, the paper is well-written and well-presented.

---

### Decision · Program_Chairs · 2023-01-20

**Decision:**

Reject

**Justification For Why Not Higher Score:**

Not much research contribution. No responses from the authors to reviews of good quality.

**Justification For Why Not Lower Score:**

Clearly written and possibly useful to practitioners.

**Metareview: Summary, Strengths And Weaknesses:**

(a) Two new heuristic methods to benefit from known future feature values in forecasting time series.

(b) Clear, sensible, well-motivated.

(c) Not enough depth. No clarity that the design of experiments was fair to alternative methods.

This paper is listed at https://jimengshi.github.io/publication/, so the submission is not anonymous.

**Summary Of Ac-Reviewer Meeting:**

No meeting.